# Heat-related mortality in U.S. state and private prisons: A case-crossover analysis

**Julianne Skarha**[1]*, **Keith Spangler**[2], **David Dosa**[3,4], **Josiah D. Rich**[1,5], **David A. Savitz**[1], **Antonella Zanobetti**[6]

**1** Department of Epidemiology, School of Public Health, Brown University, Providence, RI, United States of America, **2** Department of Environmental Health, School of Public Health, Boston University, Boston, MA, United States of America, **3** Warren Alpert Medical School, Brown University, Providence, RI, United States of America, **4** Department of Primary Care, Providence VAMC, Providence, RI, United States of America, **5** Center for Health and Justice Transformation, Providence, Rhode Island, United States of America, **6** Department of Environmental Health, Harvard T.H. Chan School of Public Health, Boston, MA, United States of America

* julianne_skarha@brown.edu

**Data Availability Statement:** Data Availability Statement: The Mortality in Correctional Institutions (MCI) dataset we used for this analysis is owned by the Bureau of Justice Statistics under

## Abstract

Rising temperatures and heatwaves increase mortality. Many of the subpopulations most vulnerable to heat-related mortality are in prisons, facilities that may exacerbate temperature exposures. Yet, there is scare literature on the impacts of heat among incarcerated populations. We analyzed data on mortality in U.S. state and private prisons from 2001–2019 linked to daily maximum temperature data for the months of June, July, and August. Using a case-crossover approach and distributed lag models, we estimated the association of increasing temperatures with total mortality, heart disease-related mortality, and suicides. We also examined the association with extreme heat and heatwaves (days above the 90th percentile for the prison location) and assessed effect modification by personal, facility, and regional characteristics. There were 12,836 deaths during summer months. The majority were male (96%) and housed in a state-operated prison (97%). A 10˚F increase was associated with a 5.2% (95% CI: 1.5%, 9.0%) increase in total mortality and a 6.7% (95% CI: -0.6%, 14.0%) increase in heart disease mortality. The association between temperature and suicides was delayed, peaking around lag 3 (exposure at three days prior death). Two- and three-day heatwaves were associated with increased total mortality of 5.5% (95% CI: 0.3%, 10.9%) and 7.4% (95% CI: 1.6%, 13.5%), respectively. The cumulative effect (lags 1–3) of an extreme heat day was associated with a 22.8% (95% CI: 3.3%, 46.0%) increase in suicides. We found the greatest increase in mortality among people ≥ 65 years old, incarcerated less than one year, held in the Northeast region, and in urban or rural counties. These findings suggest that warm temperatures are associated with increased mortality in prisons, yet this vulnerable population's risk has largely been overlooked.

## Introduction

The association between ambient heat and increased mortality risk has been studied extensively in the epidemiologic literature. Researchers have found that individual susceptibility to

the U.S. Department of Justice. It is the only national statistical collection that obtains detailed information about deaths in adult correctional facilities. Due to the sensitive nature of the data and to protect confidentiality, the data are restricted from general dissemination. The Inter-university Consortium for Political and Social Research administers access to the data upon successful approval of an Application for Use of the ICPSR Data Enclave (more information here: https://www.icpsr.umich.edu/web/pages/ICPSR/access/restricted/enclave.html). On December 2nd, 2019, our application was approved and investigators Skarha, Dosa, and Zanobetti were granted access to the MCI dataset. Data Citation: United States. Bureau of Justice Statistics. Mortality in Correctional Institutions: State Prisons, 2001-2019. Inter-university Consortium for Political and Social Research [distributor], 2021-12-16. https://doi.org/10.3886/ICPSR38035.v1.

**Funding:** This work was funded by grant F31-MD015932-01 (JS) from the National Institute on Minority Health and Health Disparities and grant R01-ES029950 (AZ) from the National Institute of Environmental Health Sciences (https://www.nimhd.nih.gov/). The funders had no role in study design, data collection and analysis, decision to publish, or preparation of the manuscript.

**Competing interests:** The authors have declared that no competing interests exist.

heat-related mortality varies. Some populations are particularly vulnerable including older adults [1], people with comorbidities including neurological or mental health disorders [1, 2], those who are socially isolated or have limited mobility [3], people who experience racism [4], and those disenfranchised by economic status [4, 5]. People who take medications, particularly psychotropic medications for mood regulation, also have an elevated risk of heat-related illness due to the potential for reduced thermoregulation capacity [1]. Many of these indicators of susceptibility are particularly prevalent in populations held inside the US incarceration system. Yet, there is scarce literature on the health impacts of heat exposure within these settings.

Due to many individuals in these settings coming from underserved communities, prison facilities are a center for complicated health problems that may be synergistically affected by temperature conditions. Approximately 43% of people held in state prisons in 2016 reported a previous diagnosis of a mental health disorder [6]. Communicable and noncommunicable diseases are also overrepresented in this setting [7]. Compared to the U.S. age-adjusted noninstitutionalized population, there is a higher prevalence of HIV, substance use disorders, diabetes, viral hepatitis, and tuberculosis in prison facilities [7]. Furthermore, aging is accelerated in incarceration settings. The current population is rapidly aging due to previous mandatory minimum policies and well as low parole rate [8]. Also, due to the excess stress imposed by incarceration, it is common for State Department of Corrections agencies to use a cutoff age of 50 years to define older adults [8]. In 2017, more than 20% of people sentenced to more than one year in a state or federal prison were age 50 or older [9].

The physical environment of prison facilities may also affect heat exposure. Overcrowded conditions can stress facility infrastructure and prevent proper temperature regulation. In 2015, the combined physical capacity of all federal prisons was being exceeded by 23% [10]. Aging prison facilities may not have adequate ventilation which not only increases the spread of communicable illnesses like COVID-19, but also can exacerbate temperature extremes [11]. Exposure to heat for people held in solitary confinement may vary depending on the size and conditions of the cell. Finally, incarcerated individuals may also not have been able to perform certain heat-adaptive behaviors due to limited access to cool water, fans, and other resources.

The U.S. has the highest incarceration rate in the world with more than 2.2 million people living behind bars daily and an estimated 10.3 million people passing through the U.S. incarceration system annually [12, 13]. However, there is a lack of epidemiologic literature on the association between elevated temperature and excess mortality among this vulnerable population. Thus, we aimed to investigate the effect of heat on mortality among people held in U.S. state and private prisons between 2001–2019.

## Methods

Mortality data came from the Bureau of Justice Statistics' Mortality in Correctional Institutions (MCI) dataset for years 2001–2019 [14]. Information about this dataset has been previous published [15]; briefly, this is the only national dataset that contains detailed information, including date, location, and cause, on the deaths of incarcerated adults in state and private prisons in the United States. Deaths by execution (or capital punishment) are excluded. Professional nosologists, or trained clinical coders, convert the cause of death to ICD-10 (international classification of diseases, 10th revision) codes. The MCI dataset has a 100% response rate among the 50 state departments of corrections [15]. Since we were interested in heat effects, we only focused on deaths that occurred in the warmest months of the year (June, July, and August). We examined all-cause or total mortality as well as two causes of death previously identified as being sensitive to heat: heart-disease related mortality (ICD-10 codes: I00–I09, I11, I13, I20–I51) and suicides [16–18].

We obtained hourly gridded temperature data with a 0.125-degree spatial resolution (approximately 12 x 12 km grid) from the North American Land Data Assimilation System (NLDAS-2) [19, 20]. We linked mortality data to the gridded temperature data based on the date of death and latitude and longitude of each prison. We used daily maximum temperature (calculated as the maximum hourly value between midnight and 11:00 PM local time) as the main exposure. We excluded 311 deaths that occurred in Alaska or Hawaii due to lack of temperature data. In order to describe exposure to heat on a continuous scale but also reflect the fact that the temperature-mortality relationship differs across climate zones, we centered the data at the prison location-specific mean summer temperature such than a value of 1˚F is equivalent to a 1 degree increase above the summer mean for that location. We also determined extreme heat events using a threshold of a maximum temperature greater than the 90[th] percentile for the respective prison location over a duration of 1, 2, or 3 consecutive days which we, respectively, defined as an extreme heat day, two-day heatwave, and three-day heatwave. Given our definition of extreme heat, the cutoff point for some prison locations was quite mild. Thus, similar to previous studies [16], we choose to exclude prisons in mild locations where the 90th percentile was less than or equal to 77˚F (25˚C) for a total number of 262 deaths excluded. Finally, we also excluded 2,711 deaths that occurred in 12 prison facilities across the U.S. that are specifically designated for medical treatment and operate similar to a hospital. A list of these 12 facilities can be found in the Supporting Information (S1 File).

The institutional review board at Brown University waived review since this does not constitute living human subjects research. We followed the Strengthening the Reporting of Observational Studies in Epidemiology (STROBE) reporting guidelines checklist for case-control studies.

## Statistical methods

We used a time-stratified case-crossover analysis to determine the effect of daily temperature and heat wave events on mortality in prisons during the period 2001–2019. We chose this study design because it captures the effect of short-term exposures (temperature) on acute outcomes (mortality) [21]. It is a variation of the matched case-control design where within each stratum an individual (or, in this case, decedent) is their own control and the temperature on the day of death is compared to the temperature on multiple reference days within the same year and month. A key advantage of this design is that it controls for confounding by season or time trends, and since each subject serves as their own control, there is no confounding by time-invariant confounders and individual characteristics such as age, gender, diet, smoking [21]. For each of our outcomes of interests (total mortality, heart disease-related mortality, and suicides), we fitted a conditional logistic regression model with a strata variable for person.

To identify the influential period of heat exposure before death, we used a distributed-lag linear model (DLM). The DLM allows us to examine the timing of the exposure response and thus the ability to determine delayed effects [22]. We modeled the lag-response using a natural cubic spline with two knots equally spaced in the log scale over a period of 14 days in order to investigate potentially long delays in the heat-mortality association. Based on these models, we determined the lags corresponding to the strongest contribution to the overall effect of temperature on mortality risk, and we computed the moving averages (the average temperature on that day and previous days) for these lags. We then tested whether the relationships between the most relevant lag and each of the mortality cause were non-linear using a natural cubic spline with two knots placed at equal spaces in the temperature range. In order to be consistent with us previously centering the data at the mean summer temperature, we used zero as the baseline temperature to model the splines. We report the results as percent change in total

mortality for a 10˚F increase above the prison-specific summer mean temperature and calculated the attributable fraction (Eq 1) and attributable number (Eq 2) of deaths to a 10˚F increase in summer temperature. We similarly fitted conditional logistic regression models for each heat variable (extreme heat day, two-day heatwave, and three-day heatwave) and calculated the associated percent change in total mortality, heart disease-related mortality, and suicides. We performed all statistical analyses in R v. 4.1.2.

$$AF_x = 1 - \exp(-\beta_x), \qquad \text{Eq 1}$$

where $\beta_x$ is the effect estimate for a 10˚F increase in summer temperature

$$AN_x = n \cdot AF_x, \qquad \text{Eq 2}$$

where $n$ is the number is the total number of deaths

## Subgroup analyses

Based on previous studies [1, 5, 16, 23, 24], as well as our own hypotheses, we examined several personal characteristics and prison characteristics as potential modifiers of the impact of heat on total mortality. We considered age at death ($\leq 44$, 45–54, 55–64, $\geq 65$) due to increasing risk mortality with age. In accordance with previous studies findings that heat exposure may have higher health impacts in climatically cool places [25, 26], we estimated effect estimates by prison location region in the U.S. (Northeast, Midwest, South, West). We include a map (S2 File) in our Supporting Information to demonstrate how we defined region. We looked at differences by security level. We obtained information on the prison security level from the Bureau of Justice Statistic's Census of State and Federal Adult Correctional Facilities [27]. This census contains facility-level data for each state correctional facility in the United States, such as the facility capacity, the number of staff, and facility security level. We include descriptions for the security level classifications in our Supporting Information (S1 Appendix). Our hypothesis was that the additional security measures in high security facilities, such as bars and steel doors, may increase the indoor facility temperature and lead to higher exposure for individuals held there. Due to minimal sample size in Super Maximum and Other classification, we focused on the three remaining categories; Low, Medium, and High. We also examined differences by length of time incarcerated before death ($< 1$ year, 1 year to 10 years, $> 10$ years) assuming that there may be an acclimating period to the prison environment as well as previous evidence on length of incarceration associated with increased risk of mortality. Finally, we used data from the 2010 U.S. Census to look at the percent rurality of the county the prison was located in as an indicator of urbanization as well as potential for the urban heat island effect which is not captured in the NLDAS-2 data [28]. We used the 2010 U.S. Census Bureau urban/rural definition to create three categories for the prison county; urban ($<10\%$ rural), mostly urban (10% - 50% rural), and rural ($>50\%$ rural) [29]. We ran conditional logistic regression models with an interaction term for the characteristic of interest and extracted the relevant P-values.

## Sensitivity analysis

We considered two other temperature indices to compare with our choice of maximum temperature as the primary exposure. We used heat index, which is computed using both humidity and temperature, and wet bulb globe temperature which combines temperature, humidity, wind speed, and solar radiation into a single index [30]. We hypothesized that these measures may better capture the way temperatures are experienced.

## Results

Our analyses included 12,836 deaths from 962 unique state or private prison facilities during summer months between 2001–2019. Heart disease-related and suicides accounted for 27% and 8% of deaths, respectively. Table 1 shows descriptive characteristics of the mortality across all prisons. Most deaths occurred among people who were male (95.5%) and held in a state-operated facility (97.4%). People aged 55 and older and people who were white accounted for 48.2% and 52.5%, respectively. Seventeen percent of deaths were among people who had been incarcerated for less than one year and 22% of deaths occurred in urban counties. High security prisons (48.2%) and the South region (47.0%) had the highest mortality within their respective categories.

Fig 1 shows the lag-response curve for the association of a 10˚F increase above the mean summer temperature and each of our outcomes of interest. The lag-response relationship was similar for total and heart disease-related mortality, with the strongest effects at lag 0 (same day exposure), decreasing but positive at lag 1, and then becoming null from lag 2 to lag 14. The shape was different for suicides with a null association at lag 0, peaking at lags 2–3 (two- or three-days prior exposure) and then becoming null again from lag 4 to 14. Based on Fig 1 we considered lag 0 for total and heart disease mortality and lag 3 for suicide to be the most predictive lag period of exposure. In our S1 Fig we present the spline between the single lag exposure defined above and each cause of death. The splines confirmed that these relationships are approximately linear.

The associations between temperature and our three outcomes are presented in Table 2 for both continuous temperature and for an extreme heat day and heatwaves. At lag 0, a 10˚F increase in temperature was associated with a 5.2% (95% CI: 1.5%, 9.0%) increase in total mortality and 6.7% (95% CI: -0.6%, 14%) increase in heart disease-related mortality. The risk estimates gradually decreased across the moving averages from lag 0–1 to lag 0–3. Conversely, we saw the risk estimates gradually increase over the moving averages for suicides, though not statistically significant; a 10˚F increase in temperature was associated with an 8.8% (95% CI: -7.0%, 25%) increase in suicide for the lag 0–3 moving average. The attributable fraction for total mortality was 4.9% which is equivalent to 635 deaths that may be attributable to each 10˚F increase above the mean summer temperature between 2001–2019.

Extreme heat was also associated with increased total mortality. An extreme heat day, two-day heatwave, and three-day heatwave were associated with a 3.5% (95% CI: -1.2%, 8.3%), 5.5% (95% CI: 0.3%, 11%), and 7.4% (95% CI: 1.6%, 14%) increase in mortality, respectively. We saw similar trends with heart disease-related mortality and suicide, though not statistically significant, but with similar effect sizes.

Due to our previous findings that temperature-suicide association may also be delayed, we further investigated the suicide lag-response relationship for an extreme heat day using a natural cubic spline. Using the cumulative function in the *dlnm* package, we found that across lags 1–3, an extreme heat day was associated with a 23% (95% CI: 3.3%, 46%) increase in suicides compared to a non-extreme heat day (Fig 2).

We found evidence of effect modification by age, length of time incarcerated, region, and urbanization (Table 3). The largest increases in total mortality were among people 65 years and older, incarcerated less than one year or more than 10 years, living in the Northeast region, and held in an urban or rural county (when compared to a mostly urban county). For example, during a two-day heatwave, mortality increased by 21% (95% CI: 6.2%, 37%) in the Northeast compared to 0.8% (95% CI: -10%, 13%) in the Midwest (interaction p-value = 0.04), 1.3% (95% CI: -6.1%, 9.3%) in the South (interaction p-value = 0.02), and 8.6% (95% CI: -2.5%, 21%) in the West (interaction p-value = 0.21). Generally, these effect modification trends

**Table 1. Descriptive characteristics of mortality during summer months while under custody of a state or private prison from 2001–2019[a-f].**

| Characteristic | Population |
|---|---|
| | **N = 12,836** |
| **Sex, N (%)** | |
| Female | 583 (4.5) |
| Male | 12,253 (95.5) |
| **Age, N (%)** | |
| ≤44 | 3,263 (25.4) |
| 45–54 | 3,380 (26.3) |
| 55–64 | 3,343 (26.1) |
| ≥65 | 2,837 (22.1) |
| Missing | 13 (0.1) |
| **Race, N (%)** | |
| Black | 4,487 (35.0) |
| Hispanic | 1,358 (10.6) |
| White | 6,742 (52.5) |
| Other[b] | 226 (1.8) |
| Missing | 23 (0.1) |
| **Length of incarceration prior to death, N (%)** | |
| < 1 year | 2,193 (17.1) |
| 1 year to 10 years | 5,818 (45.3) |
| > 10 years | 4,763 (37.1) |
| Missing | 62 (0.5) |
| **Facility Operator, N (%)** | |
| State | 12,502 (97.4) |
| Private | 310 (2.4) |
| Missing | 24 (0.2) |
| **Security Level of Facility, N (%)** | |
| Super maximum | 209 (1.6) |
| High | 6,183 (48.2) |
| Medium | 4,910 (38.3) |
| Low | 1,147 (8.9) |
| Other | 387 (3.0) |
| **Region** | |
| Northeast[c] | 1,889 (14.7) |
| Midwest[d] | 2,442 (19.0) |
| South[e] | 6,032 (47.0) |
| West[f] | 2,473 (19.3) |
| **Urbanization** | |
| Urban | 2,825 (22.0) |
| Mostly Urban | 5,410 (42.1) |
| Rural | 4,435 (34.6) |
| Missing | 166 (1.3) |

[a]June, July, and August are used as summer months

[b]Includes persons who were identified as Asian, Native Hawaiian, Other Pacific Islander, American Indian, Alaska Native, or persons of two or more races

[c]Northeast = CT, ME, MA, NH, NJ, NY, PA, RI, VT

[d]Midwest = IL, IN, IA, KS, MI, MN, MO, NE, ND, OH, SD, WI

[e]South = AL, AR, DE, DC, FL, GA, KY, LA, MD, MS, NC, OK, SC, TN, TX, VA, WV

[f]West = AZ, CA, CO, ID, MT, NY, NM, OR, UT, WA, WY

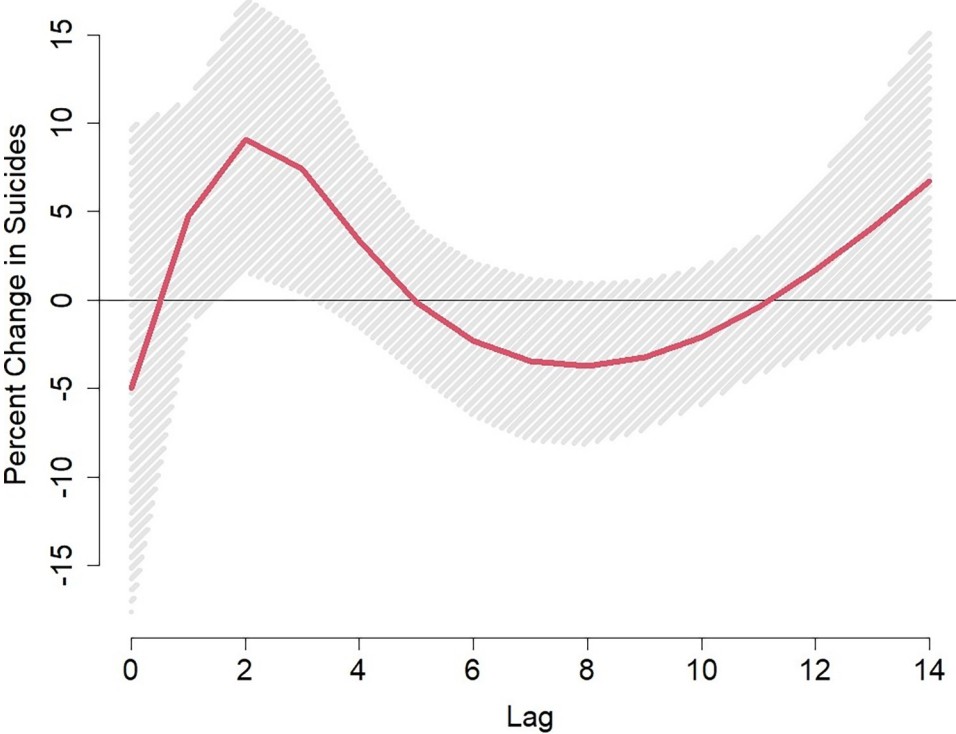

**Fig 1. Lag-response curves representing the percent change in total mortality, heart disease-related mortality, and suicides associated with a 10˚F degree increase in mean temperature above the prison-specific summer average at each lag in U.S. prisons between 2001–2019.**

carried across the extreme heat and heatwave days, but not for continuous temperature, except for urbanization. While we found some evidence of difference by security level of facility in the association between total mortality and a 10˚F increase in temperature (Low: 13% (95% CI: 0.0%, 26%) vs Medium: 1.7% (95% CI: -4.6%, 7.9%), interaction p-value = 0.12), this trend was not present over extreme heat days (interaction p-values ≥ 0.40).

**Table 2. Percent change in total mortality, heart disease-related mortality, and suicides associated to a 10˚F increase in temperature averaged over a period up to three days, and extreme heat and heatwave days during summer months[a].**

| | Total Mortality, | Heart Disease-Related Mortality | Suicide, |
|---|---|---|---|
| | % (95% CI) | % (95% CI) | % (95% CI) |
| **Sample Size** | **12,836** | **3,463** | **1,016** |
| **Continuous heat** | | | |
| Lag 0 | 5.2 (1.5, 9.0) | 6.7 (-0.6, 14) | 4.8 (-8.1, 18) |
| Lag 0–1 | 4.8 (0.7, 9.0) | 5.9 (-2.0, 14) | 5.6 (-8.4, 20) |
| Lag 0–2 | 3.7 (-0.6, 8.0) | 5.3 (-3.2, 14) | 6.9 (-8.0, 22) |
| Lag 0–3 | 3.2 (-1.4, 8.0) | 4.8 (-4.2, 14) | 8.8 (-7.0, 25) |
| **Extreme heat & heatwaves** | | | |
| Extreme heat day | 3.5 (-1.2, 8.3) | 0.9 (-7.7, 10) | 2.2 (-13.2, 20) |
| Two-day heatwave | 5.5 (0.3, 11) | 5.8 (-4.0, 17) | 4.5 (-12.6, 25) |
| Three-day heatwave | 7.4 (1.6, 14) | 7.6 (-3.3, 20) | 15 (-5.6, 40) |

[a]June, July, and August are used as summer months

We did not find evidence that different temperature indices would better capture the relationship between heat and mortality among this population (S1 Table).

## Discussion

In this case-crossover study of U.S. state and private prisons, we found an association between increasing continuous temperature, extreme heat, heatwaves days and mortality, with marked

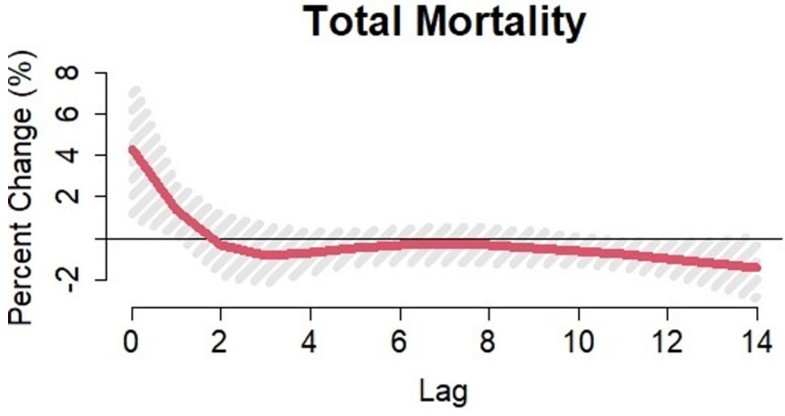

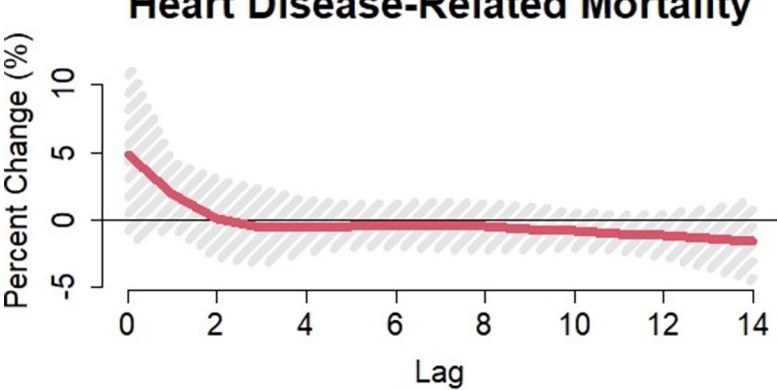

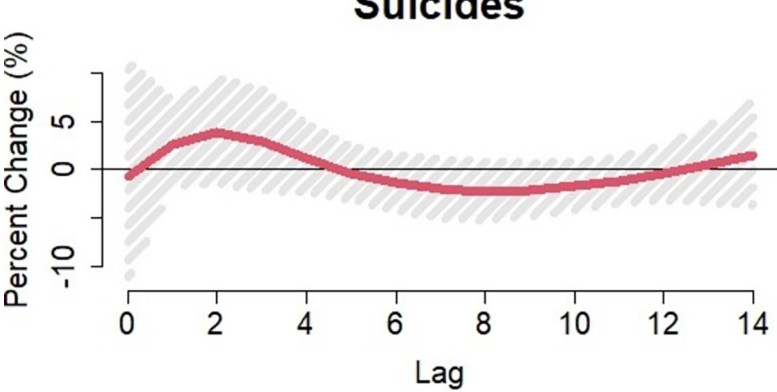

**Fig 2. Lag-response curves for the percent change in suicides associated to an extreme heat day in U.S. prisons between 2001 – 2019[a].** [a]An extreme heat day is defined as a daily maximum temperature above the 90th percentile for the respective prison location.

**Table 3. Percent change in mortality with 95% confident intervals associated to temperatures in summer months using an interaction term for personal, facility, and regional characteristics in US prisons from 2001–2019[a,b,c,d].**

| Change in total mortality at: | 10˚F increase in temperature, % (95% CI) | p-value | Extreme Heat Day, % (95% CI) | p-value | Two-day Heatwave, % (95% CI) | p-value | Three-day Heatwave, % (95% CI) | p-value |
|---|---|---|---|---|---|---|---|---|
| Age Group | | | | | | | | |
| ≤ 44 | 6.1 (-1.8, 14) | Ref | 0.6 (-8.2, 10) | Ref | -2.0 (-11, 8.4) | Ref | -0.6 (-11, 11) | Ref |
| 45–54 | 4.1 (-3.6, 12) | 0.72 | 1.3 (-7.5, 11) | 0.92 | 7.0 (-3, 18) | 0.22 | 10 (-1.3, 23) | 0.20 |
| 55–64 | 5.8 (-2.1, 14) | 0.95 | 7.5 (-1.7, 18) | 0.31 | 7.7 (-2.4, 19) | 0.19 | 6.2 (-4.7, 18) | 0.41 |
| ≥ 65 | 3.0 (-5.6, 12) | 0.60 | 4.6 (-5, 15) | 0.56 | 11 (-0.3, 23) | 0.10 | 16 (3.5, 31) | 0.06 |
| Time Incarcerated | | | | | | | | |
| < 1 year | 8.3 (-1.2, 18) | Ref | 11 (-0.2, 24) | Ref | 16 (3.4, 31) | Ref | 14 (0.1, 30) | Ref |
| 1 year to 10 years | 3.0 (-3.0, 9.0) | 0.36 | -2.8 (-9.3, 4.1) | 0.04 | -2.7 (-9.8, 5.0) | 0.01 | 0.4 (-7.6, 9.2) | 0.11 |
| > 10 years | 5.8 (-0.8, 13) | 0.67 | 8.2 (0.4, 17) | 0.68 | 12 (3.3, 22) | 0.61 | 14 (4.2, 25) | 1.00 |
| Security Level | | | | | | | | |
| Low | 13 (0.0, 26) | Ref | 8.1 (-6.9, 26) | Ref | 13 (-4.2, 33) | Ref | 5.6 (-12, 27) | Ref |
| Medium | 1.7 (-4.6, 7.9) | 0.12 | 1.8 (-5.5, 9.7) | 0.48 | 5.7 (-2.6, 15) | 0.48 | 7.9 (-1.4, 18) | 0.84 |
| High | 6.1 (0.1, 12) | 0.35 | 4 (-2.7, 11) | 0.64 | 4.4 (-2.9, 12) | 0.40 | 9.3 (0.8, 18) | 0.74 |
| Region | | | | | | | | |
| Northeast | 11 (0.6, 21) | Ref | 11 (-1.0, 25) | Ref | 21 (6.2, 37) | Ref | 23 (6.6, 43) | Ref |
| Midwest | 2.3 (-5.6, 10) | 0.20 | 4.1 (-6.2, 16) | 0.41 | 0.8 (-10.2, 13) | 0.04 | 2.7 (-9.7, 17) | 0.07 |
| South | 2.6 (-4.4, 9.6) | 0.20 | -2.2 (-8.8, 4.9) | 0.06 | 1.3 (-6.1, 9.3) | 0.02 | 4.9 (-3.4, 14) | 0.06 |
| West | 6.4 (-1.6, 15) | 0.52 | 9.4 (-1.0, 21) | 0.83 | 8.6 (-2.5, 21) | 0.21 | 7.0 (-5.0, 21) | 0.14 |
| Urbanization | | | | | | | | |
| Urban | 10 (1.4, 19) | Ref | 11 (0.9, 22) | Ref | 14 (2.9, 27) | Ref | 12 (0.1, 26) | Ref |
| Mostly Urban | 0.0 (-5.8, 5.9) | 0.06 | -3.9 (-10, 3.1) | 0.02 | -4.9 (-12, 2.8) | 0.01 | -0.6 (-8.7, 8.3) | 0.10 |
| Rural | 8.6 (1.2, 16) | 0.79 | 9.0 (0.6, 18) | 0.77 | 15 (5.3, 25) | 0.94 | 15 (4.6, 27) | 0.74 |

[a]P-values should be interpreted as whether there is a difference between the percent change in mortality of a sub group compared to the reference (Ref) group

[b]June, July, and August are used as summer months

[c]More specifically, 10˚F increase in maximum temperature above the prison-specific summer mean temperature

[d]An extreme heat day is defined as a daily maximum temperature above the 90th percentile for the respective prison location while a two-day heatwave same day and previous day about the 90th percentile and a three-day heatwave: same day and previous two days above the 90th percentile

increases for heart disease-related mortality and suicide. We also saw consistent trends for effect modification by certain personal and prison location characteristics, namely among people ≥ 65 years old, incarcerated less than one year, held in the Northeast region, and in urban or rural counties. To our knowledge, no other epidemiological study has reported on heat-related increases in mortality among incarcerated populations.

Many of our findings are consistent with previous studies modeling heat and mortality outcomes, including our lag-response relationships. In a meta-analysis of 50 US cities, Medina-Ramon and Schwartz [16] found both total mortality and myocardial infarction morality peaked at lag 0 and decreased afterwards. Sheridan and colleagues [31] similar looked at the largest 51 metropolitan areas in the US and reported the greatest increase in mortality from same-day exposure. However, while we found that suicides peaked 2–3 days after exposure, other studies have reported increased risk of suicide from same-day exposure. Kim and colleagues [17] performed a multi-country study combining 341 locations and generally found an association at lag 0 or lag 1 only. A study from Brazil similarly reported an increase in suicides associated with same day temperature only [18]. Still, suicides are complex events and may look particularly different in prison settings, even in terms of when they are recorded. Thus, lagged effects may be an important part of the association between heat and suicides in prison.

For risk differences, we found that by age groups, the effect of extreme heat days was greatest among people 65 years and older and this has been similarly reported in other highly studied heatwave events, such as the 1995 Chicago heat wave and the 2003 European heatwave [3, 32]. For time incarcerated, we found heat-related mortality increased more among people who had been incarcerated less than one year. Previous evidence indicates that mortality risks may be lower among people who are heat-acclimated. In a study of heat-related deaths in Maricopa County, Arizona, researchers found that non-Arizona residents were 5 times more likely to have an outdoor heat-related death [33]. Potentially, a similar phenomenon is occurring among people who become acclimated to the prison environment. We observed mortality increases among people incarcerated for 10+ years compared to people incarcerated for 1–10 years prior to death. This may be partly explained by increasing evidence that the length of incarceration reduces the life span [23]. We saw heterogeneity by U.S. region, with the association between heat and mortality being highest in the Northeast and minimal in the Midwest, South, and West. This is consistent with other research findings that hot days are less deadly in climatically warm places [26]. When looking at effect modification by urbanization, we found the largest increases in very urban counties and rural counties with no increase in mostly urban areas. Interestingly, this "u-shape" effect has been reported elsewhere in the epidemiologic literature such that the risk of heat-related mortality is high is rural locations but also increases with population density [5, 16, 24]. Some of the proposed hypotheses for this relationship include hospital access, proportion of families living in poverty, and proportion of elderly residents. However, not all of these apply to prison settings and further investigation into this association would provide deeper insight into urbanization and prisons.

A key difference in our study, compared to others across the United States, is the size of our effect estimates. Medina-Ramon and Schwartz [16] found an extreme temperature day (day $\geq 99^{th}$ percentile) increased mortality by 3.85% (2.54%, 5.18%) at lag 0. Sheridan et al. [31] looked at relative extreme heat events (day $> 85^{th}$ percentile) and found a 1.8% (1.0%, 2.5%) increase in mortality. We found that an extreme heat day (day $> 90^{th}$ percentile) was associated with a 3.5% (95% CI: -1.2, 8.3) increase in mortality. Anderson and Bell [26] report a heatwave (two days $\geq 95^{th}$ percentile) was associated with 3.74 (2.29%, 5.22%) rise in mortality. We found that a two-day heatwave was associated with 5.5% (95% CI: 0.3%, 11%) increase in mortality. Our effect estimates are generally higher. Part of this difference may be due to definitions (as we looked at days $> 90^{th}$ percentile) but part of this may due to the prison setting itself. The population incarcerated in the United States in not representative of the general population. In 2012, 83% of people held in state prison facilities were under the age of 50 years old [7]. Yet, adults held in prison are 3.4 times more likely to report heart-related health problems and 1.5 times more likely to report diabetes when compared to a standardized noninstitutionalized US population [7]. More research is needed to investigate how the structural prison environment itself is affecting health outcomes, including by exacerbating temperature exposures.

This study is not without limitations. We did not have information about which prison facilities or units within the facilities were air conditioned, which had been shown to be protective for heat-related health effects. This averaging of effects for those with AC likely led us to underestimate the association between heat and mortality in facilities without AC. Additionally, when determining effect measure modification, we assumed similar relationships within each characteristic category (mainly, linear and a moving average using lag 0–1 as the most relevant period). However, these assumptions may not hold for the relationship between heat and mortality within some of these subcategories. Due to our data being spread over 962 facilities across a 19-year period, we were limited in our ability to look at more extreme events, such as days $> 95^{th}$ percentile, which may be a better measure for extreme heat exposure. Finally,

we had limited information on the type of conditions someone was held in before they died. Were they being held in solitary confinement and for how long? Were they working outside? Were they housed in a geriatric-care unit? More granular data would allow us to better capture the true effect estimate as well as provide insight into important effect modifiers. However, to our knowledge, these types of data do not currently exist.

## Conclusion

This study provides some of the first epidemiologic evidence that people who are incarcerated in prison facilities may be particularly susceptible to heat-related mortality. Yet, we do not fully understand the unique ways this population may experience heat that are distinctive to this environment, such as through work conditions, solitary confinement, or restriction of resources. Furthermore, exposure may look different in other carceral settings, such as jails or immigrant detention facilities. Finally, mortality is just one health outcome of interest and does not capture the full range of ways heat can affect health. Having quality, valid, and reliable health data that is publicly available will be imperative to further investigating temperature exposures in carceral settings.

## Supporting information

**S1 Fig. Association between temperature above prison-specific summer mean temperature and total mortality, heart disease-related mortality, and suicide at the most relevant lag period using a natural cubic spline with two knots.**
(DOCX)

**S1 Table. Using different temperature metrics with the moving average (Lag01) to model the percent change in total mortality during summer months in U.S. prisons.**
(DOCX)

**S1 File. List of medical prison facilities removed from analyses.**
(DOCX)

**S2 File. Regional breakdown of the United States into Northeast, Midwest, South, and West.**
(DOCX)

**S1 Appendix. Description of prison facility security level classification.**
(DOCX)

## Author Contributions

**Conceptualization:** Julianne Skarha, Keith Spangler, David Dosa, Josiah D. Rich, David A. Savitz, Antonella Zanobetti.

**Data curation:** Julianne Skarha, Keith Spangler.

**Formal analysis:** Julianne Skarha.

**Funding acquisition:** Julianne Skarha.

**Investigation:** Julianne Skarha.

**Methodology:** Julianne Skarha.

**Supervision:** David Dosa, Josiah D. Rich, David A. Savitz, Antonella Zanobetti.

**Validation:** Keith Spangler, Antonella Zanobetti.

**Writing – original draft:** Julianne Skarha.

**Writing – review & editing:** Julianne Skarha, Keith Spangler, David Dosa, Josiah D. Rich, David A. Savitz, Antonella Zanobetti.

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
