## [Decision Letter · Decision Letter 0]

28 Oct 2022

PONE-D-22-19497Heat-related mortality in U.S. state and private prisons: a case-crossover analysisPLOS ONE

Dear Dr. Skarha,

Thank you for submitting your manuscript to PLOS ONE. After careful consideration, we feel that it has merit but does not fully meet PLOS ONE’s publication criteria as it currently stands. Therefore, we invite you to submit a revised version of the manuscript that addresses the points raised during the review process.

Reviewers have now commented on your paper. Both reviewers found your manuscript very relevant, and are willing to consider a revised version. Please make sure to address point-by-point their recommendations in a revised version. Please find the attached file for Reviewer 1's comments.

We look forward to receiving your revised manuscript.

Kind regards,

Byung Chul Chun, M.D., MPH, Ph.D

Academic Editor

PLOS ONE

Journal Requirements:

Reviewers' comments:

Reviewer's Responses to Questions

**Comments to the Author**

1. Is the manuscript technically sound, and do the data support the conclusions?

Reviewer #1: Yes

Reviewer #2: Yes

2. Has the statistical analysis been performed appropriately and rigorously? 

Reviewer #1: Yes

Reviewer #2: Yes

3. Have the authors made all data underlying the findings in their manuscript fully available?

Reviewer #1: No

Reviewer #2: No

4. Is the manuscript presented in an intelligible fashion and written in standard English?

Reviewer #1: Yes

Reviewer #2: Yes

5. Review Comments to the Author

Reviewer #1: The paper tried to understand the impact of extreme heats on the incarcerated populations using case-crossover analysis. Given the limited number of previous papers focusing on those people, I strongly believe this paper can provide some valuable insights and information. Overall, I really enjoyed reading this manuscript. I raised some major and minor comments. Most of my comments were for clarification. I believe this paper fits well with the PLOS One, and this manuscript can be published after some revisions. Please see the file attached.

Reviewer #2: This manuscript is an analysis of the heat-related mortality among incarcerated population in prison in the United States and provides valuable insight. It is also worthwhile in that there are little similar studies conducted on this subpopulation. However, there are some parts need to be made clear:

1. Line 96 Deaths in federal prison facilities are collected elsewhere.

Please specify the source of information of deaths in federal prison facilities.

2. Line 103 heart-disease related mortality (ICD-10 codes: I00–I09, I11, I13, I20–I51) and suicides.

- This sentence requires references for the diseases and suicide.

3. Line 116-118 We excluded prisons in mild climate locations where the 90th percentile was less than or equal to 77°F (number of deaths = 262).

- Please provide reference(s) for the threshold of 77°F for exclusion.

4. Line 158-159 security level, Line 399 Table 1, Line 427 Table 3

- You need to explain the classification of the security levels. What is the major difference between “low”, “middle”, “high” and “super maximum”?

- Clarify your source of reference on the security levels of the facilities.

- Table 1: What does you mean 387 “Others” in security level?

- Table 3: There were only 3 levels of security in Table 3. How did you treat the “super maximum” and “other” security levels in the subgroup analysis?

5. Line 399 Table 1, Line 427 Table 3

- Are there any specific reasons to exclude the Race and Facility Operator in the subgroup analysis? If you present the differences by the Race and Facility Operator, this article will be more informative, even if not statistically significant.

6. PLOS authors have the option to publish the peer review history of their article (what does this mean?). If published, this will include your full peer review and any attached files.

Reviewer #1: **Yes: **Jihoon Jung

Reviewer #2: No

---

## [Author Response · Author response to Decision Letter 0]

12 Dec 2022

EDITORIAL REQUIREMENTS

Our manuscript is now formatted to the PLOS ONE style requirements. 

We have updated our cover letter with the following information: The Mortality in Correctional Institutions (MCI) dataset we used for this analysis is owned by the Bureau of Justice Statistics under the U.S. Department of Justice. It is the only national statistical collection that obtains detailed information about deaths in adult correctional facilities. Due to the sensitive nature of the data and to protect confidentiality, the data are restricted from general dissemination. The Inter-university Consortium for Political and Social Research administers access to the data upon successful approval of an Application for Use of the ICPSR Data Enclave (more information here: https://www.icpsr.umich.edu/web/pages/ICPSR/access/restricted/enclave.html). On December 2nd, 2019, our application was approved and investigators Skarha, Dosa, and Zanobetti were granted access to the MCI dataset. 

Data Citation: United States. Bureau of Justice Statistics. Mortality in Correctional Institutions: State Prisons, 2001-2019. Inter-university Consortium for Political and Social Research [distributor], 2021-12-16. https://doi.org/10.3886/ICPSR38035.v1

We have followed the Supporting Information guidelines and include captions at the end of the manuscript as well as have updated any in-text citations. 

Reviewer 1

Remarks to the Author: The paper tried to understand the impact of extreme heats on the incarcerated populations using case-crossover analysis. Given the limited number of previous papers focusing on those people, I strongly believe this paper can provide some valuable insights and information. Overall, I really enjoyed reading this manuscript. I raised some major and minor comments. Most of my comments were for clarification. I believe this paper fits well with the PLOS One, and this manuscript can be published after some revisions.

Comments: 

1. Line 158: “We considered age at death (44, 45 - 54, 55 - 64, 65), security level of prison facility (low, medium, high), and prison location region in the U.S. (Northeast, Midwest, South, West)” 

Please provide any justifications in the introduction why you looked at the impact of security level of prison facility and prison location region in the US. Did you assume that security level is a proxy for stress? I cannot find any reasons why you looked at these variables in the manuscript.

The Reviewer makes a good point and we have expanded to the manuscript to describe our reasoning for looking at prison location region and security level. 

Line 161: In accordance with previous studies findings that heat exposure may have higher health impacts in climatically cool places [25,26], we estimated effect estimates by prison location region in the U.S. (Northeast, Midwest, South, West). We include a map (S1 Map) in our Supporting Information to demonstrate how we defined region. We looked at differences by security level. We obtained information on the prison security level from the Bureau of Justice Statistic’s Census of State and Federal Adult Correctional Facilities[27]. This census contains facility-level data for each state correctional facility in the United States, such as the facility capacity, the number of staff, and facility security level. We include descriptions for the security level classifications in our Supporting Information (S1 Appendix). Our hypothesis was that the additional security measures in high security facilities, such as bars and steel doors, may increase the indoor facility temperature and lead to higher exposure for individuals held there.

2. Can you also provide any map showing your region classification (Northeast, Midwest, South, West)? I know you added some information in the table. But, some audiences outside of the US would have a very limited understanding on this research area. I think showing a map is one of the most intuitive ways.

We appreciate the Reviewer’s point and have added a map showing the region classifications to our Supporting Information and reference it in the manuscript. 

Line 164: We include a map (S1 Map) in our Supporting Information to demonstrate how we defined region.

3. Line 116: “We excluded prisons in mild climate locations where the 90th percentile was less than or equal to 77°F (number of deaths = 262)” 

Do you have any specific reasons for selecting 77°F as a threshold.

Because our definition for an extreme heat day was the 90th percentile of the maximum temperature for a given prison location, some of the cutoff points defined as extreme temperature for some prison locations were in fact quite mild. Thus, we chose, as previous studies have done, to use a cutoff point which we selected as 25°C (or 77°F). We expanded in the manuscript to further explain this choice. 

Line 110: Given our definition of extreme heat, the cutoff point for some prison locations was quite mild. Thus, similar to previous studies [16], we chose to exclude prisons in mild locations where the 90th percentile was less than or equal to 77°F (25°C) for a total number of 262 deaths excluded. 

16. Medina-Ramon M, Schwartz J. Temperature, temperature extremes, and mortality: a study of acclimatisation and effect modification in 50 US cities. Occupational and Environmental Medicine. 2007;64: 827–833. doi:10.1136/oem.2007.033175

4. Line 166 “We created three categories based on the percentage of the county classified as rural; urban (10% rural), mostly urban (10% - 50% rural), and rural (50% rural).” 

Did you arbitrarily select these categories? Maybe this could be redundant. But, for those who think that NLDAS-2 data already captured urban heat island effects, maybe I would add a sentence showing that urban heat island effects are not captured by NLDAS-2 (Crosson 2020), so we wanted to separately check the impact of urban heat island effects.

Crosson, W. L., Al-Hamdan, M. Z., & Insaf, T. Z. (2020). Downscaling NLDAS-2 daily

maximum air temperatures using MODIS land surface temperatures. Plos one, 15(1),

e0227480.

We chose these categories based on the U.S. Census Bureau definition of a 50% cutoff to delineate between mostly rural and mostly urban for the 2010 U.S. Census. We now reference the paper in which this is defined and expand further in our manuscript to offer this explanation. We thank the Reviewers for the reference that urban heat island effects are not captured by the NLDAS-2 and expand upon this in our manuscript as well as reference this paper. 

Line 176: Finally, we used data from the 2010 U.S. census to look at the percent rurality of the county the prison was located in as an indicator of urbanization as well as potential for the urban heat island effect which is not captured in the NLDAS-2 data. 

Line 179: We used the 2010 U.S. Census Bureau urban/rural definition to create three categories for the prison county; urban (<10% rural), mostly urban (10% - 50% rural), and rural (>50% rural) [27]. 

27.Ratcliffe M, Burd C, Holder K, Fields A. Defining Rural at the U.S. Census Bureau. Washington, DC: U.S. Census Bureau; 2016. Report No.: ACSGEO-1. Available: https://www.census.gov/content/dam/Census/library/publications/2016/acs/acsgeo-1.pdf

5. Line 185 “High security prisons and the South region had the highest mortality within their respective categories.” 

I would add the actual values (For consistency). “High security prisons (48.2%) and the South region (47.0%) had the highest mortality within their respective categories.”

We appreciate the Reviewer’s point and have updated the manuscript with the actual values. 

Line 198: High security prisons (48.2%) and the South region (47.0%) had the highest mortality within their respective categories.

6. Lines 209-212: This sentence sounds awkward: “Due to our previous findings that temperature-suicide association may also be delayed, Figure 2 shows the suicide lag-response relationship for an extreme heat day using a natural cubic spline.” (Maybe) “Due to our previous findings that temperature-suicide association may also be delayed, we further investigated the suicide lag-response relationship for an extreme heat day using a natural cubic spline.” And more importantly, I cannot see any reason for doing this analysis (maybe I am wrong). I think Figure 1 and Figure 2 contain the same information. Is there any additional information? And where does the number 23% come from? Is this a cumulative percent across lags 1-3?

We agree with the Reviewer about the phrasing and have updated this in the manuscript. Figure 1 is for a 10°F increase in the mean temperature while Figure 3 is for an extreme heat day. The 23% increase in risk of suicide is a cumulative estimate using a function in the dlnm package that predicts the incremental cumulative associations along the specified lag period. We updated language in the manuscript to better explain this. 

Line 248: Due to our previous findings that temperature-suicide association may also be delayed, we further investigated the suicide lag-response relationship for an extreme heat day using a natural cubic spline. 

Line 250: Using the cumulative function in the dlnm package, we found that across lags 1-3, an extreme heat day was associated with a 23% (95% CI: 3.3%, 46%) increase in suicides compared to a non-extreme heat day (Figure 2).

7. From Figure 1 and Table 1, I think most of the impacts are not statistically significant. Only total mortality lag 0 and lag 0-1and total mortality two-day heatwave three-day heatwave are statistically significant. I know you mentioned that some results are not statistically significant (Line 207-208). But, this could mislead audiences that other results excepts them in Lines 207-208 are statistically significant. So, I recommend you first talk about statistically significant results and then introduce other results which are not statistically significant (although not significant, we found some evidence showing ~~~).

We focused more on the trend of the estimates, but we have updated our language to acknowledge that although there is an upward trend, the individual risk estimates are not statistically significant. 

Line 232: Conversely, we saw the risk estimates gradually increase over the moving averages for suicides, though not statistically significant; a 10°F increase in temperature was associated with an 8.8% (95% CI: -7.0%, 25%) increase in suicide for the lag 0-3 moving average.

8. Table 2. I think most of the relationships are not statistically significant. I think you may need to mention that these results are not statistically significant (e.g. Although mostly not significant, we found some evidence~~ (Line 213~). Some readers may think that all of these results are statistically significant.

Similar to the previous response, we have updated our language to now describe these results as not statistically significant. 

Line 246: We saw similar trends with heart disease-related mortality and suicide, though not statistically significant, but with similar effect sizes.

Reviewer 2

Remarks to the Author: This manuscript is an analysis of the heat-related mortality among incarcerated population in prison in the United States and provides valuable insight. It is also worthwhile in that there are little similar studies conducted on this subpopulation. However, there are some parts need to be made clear:

Comments

1. Line 96: “Deaths in federal prison facilities are collected elsewhere.” 

Please specify the source of information of deaths in federal prison facilities.

Data on deaths that occur in federal prisons are housed by the Bureau of Prisons, a separate agency from the Bureau of Justice Statistics. Since our data came from the BJS, we only had data on deaths in state-operated and state contracted privately-operated facilities. However, approximately 85% of people incarcerated in prison are being held in state-operator facility. We believe mentioning the federal prisons may create confusion for the reader so we have removed this sentence from the manuscript. 

2. Line 103: “heart-disease related mortality (ICD-10 codes: I00–I09, I11, I13, I20–I51) and suicides.” 

This sentence requires references for the diseases and suicide.

We appreciate the Reviewer catching this and we have now updated this sentence with the appropriate references. 

3. Line 116-118: “We excluded prisons in mild climate locations where the 90th percentile was less than or equal to 77°F (number of deaths = 262).” 

Please provide reference(s) for the threshold of 77°F for exclusion.

Because our definition for an extreme heat day was the 90th percentile of the maximum temperature for a given prison location, some of the cutoff points defined as extreme temperature for some prison locations were in fact quite mild. Thus, we chose, as previous studies have done, to use a cutoff point which we selected as 25°C (or 77°F). We expanded in the manuscript to further explain this choice. 

Line 110: Given our definition of extreme heat, the cutoff point for some prison locations was quite mild. Thus, similar to previous studies [16], we choose to exclude prisons in mild locations where the 90th percentile was less than or equal to 77°F (25°C) for a total number of 262 deaths excluded. 

16. Medina-Ramon M, Schwartz J. Temperature, temperature extremes, and mortality: a study of acclimatisation and effect modification in 50 US cities. Occupational and Environmental Medicine. 2007;64: 827–833. doi:10.1136/oem.2007.033175

4. Line 158-159 security level, Line 399 Table 1, Line 427 Table 3

a) You need to explain the classification of the security levels. What is the major difference between “low”, “middle”, “high” and “super maximum”?

We now include the definitions of the security level classification as defined in the Bureau of Justice Statistic’s Census of State and Federal Adult Correctional Facilities in our Supporting Information materials. 

S1 Appendix. Description of prison facility security level classification. 

Super maximum, maximum/close/high security is characterized by walls or double-fence perimeters, armed towers, or armed patrols. Cell housing is isolated in one of two ways: within a cell block so that a prisoner escaping from a cell is confined within the building or by double security from the perimeter by bars, steel doors, or other hardware. All entry or exit is via trap gate or sally port.

Medium security is characterized by a single or double-fenced perimeter with armed coverage by towers or patrols. Housing units are cells, rooms, or dormitories. Dormitories are living units designed or modified to accommodate 12 or more persons. All entry or exit is via trap gate or sally port.

Minimum or low security is characterized by a fenced or “posted” perimeter. Cell housing units are rooms or dormitories. Normal entry and exit are under visual surveillance.

b) Clarify your source of reference on the security levels of the facilities.

We expanded the description of the Bureau of Justice Statistic’s Censes of State and Federal Adult Correctional Facilities.

Line 165: We obtained information on the prison security level from the Bureau of Justice Statistic’s Censes of State and Federal Adult Correctional Facilities. This census contains facility-level data for each state correctional facility in the United States, such as the facility capacity, the number of staff, and facility security level. We include descriptions for the security level classifications in our Supporting Information (S1 Appendix). 

c) Table 1: What does you mean 387 “Others” in security level?

The Other classification in security level can refer to half-way houses or administrative facilities that hold individuals temporarily. Due to the minimal sample size, it was not meaningful to run analyses in the “Other” security classification group. 

d) Table 3: There were only 3 levels of security in Table 3. How did you treat the “super maximum” and “other” security levels in the subgroup analysis?

Due to the minimal sample sizes, it was not meaningful to run analyses using the subgroups “super maximum” and “other” and so they were excluded from the security level analysis. The remaining levels were “low”, “medium”, and “high”. We have added a line in the manuscript to clarify this choice. 

Line 172: Due to minimal sample size in Super Maximum and Other classification, we focused on the three remaining categories; Low, Medium, and High.

5. Line 399 Table 1, Line 427 Table 3

Are there any specific reasons to exclude the Race and Facility Operator in the subgroup analysis? If you present the differences by the Race and Facility Operator, this article will be more informative, even if not statistically significant.

Due to the minimal sample size in the private prison subgroup for Facility Operator (N = 310), it was not meaningful to look at effect modification. With regards to Race, we wanted to be thoughtful about how race was measured in this setting, which was not self-reported. Various correctional professionals, ranging from prison administrators to correctional medical and officer staff, may complete the State Prison Inmate Death Report that records the decedent’s race. We hope to devote future research to this topic area once we have a clearer understanding of how to interpret this assignment.

---

## [Decision Letter · Decision Letter 1]

23 Jan 2023

Heat-related mortality in U.S. state and private prisons: a case-crossover analysis

PONE-D-22-19497R1

Dear Dr. Skarha,

We’re pleased to inform you that your manuscript has been judged scientifically suitable for publication and will be formally accepted for publication once it meets all outstanding technical requirements.

Kind regards,

Byung Chul Chun, M.D., MPH, Ph.D

Academic Editor

PLOS ONE

Additional Editor Comments (optional):

Reviewers' comments:

Reviewer's Responses to Questions

**Comments to the Author**

1. If the authors have adequately addressed your comments raised in a previous round of review and you feel that this manuscript is now acceptable for publication, you may indicate that here to bypass the “Comments to the Author” section, enter your conflict of interest statement in the “Confidential to Editor” section, and submit your "Accept" recommendation.

Reviewer #1: All comments have been addressed

Reviewer #2: All comments have been addressed

2. Is the manuscript technically sound, and do the data support the conclusions?

Reviewer #1: Yes

Reviewer #2: Yes

3. Has the statistical analysis been performed appropriately and rigorously? 

Reviewer #1: Yes

Reviewer #2: Yes

4. Have the authors made all data underlying the findings in their manuscript fully available?

Reviewer #1: (No Response)

Reviewer #2: No

5. Is the manuscript presented in an intelligible fashion and written in standard English?

Reviewer #1: Yes

Reviewer #2: Yes

6. Review Comments to the Author

Reviewer #1: Every answer was satisfactory. Thanks for working really hard on my comments. I hope my comments were helpful.

Reviewer #2: Thanks for your revision. It may be needed to compare the heat-associated mortality of the general population with that of the correctional facility population by the region. If the age and/or ethnicity distribution of the population within the correctional facility differs by region where the correctional facility is located, it may be helpful to consider this in the analysis.

7. PLOS authors have the option to publish the peer review history of their article (what does this mean?). If published, this will include your full peer review and any attached files.

Reviewer #1: **Yes: **Jihoon Jung

Reviewer #2: No

---

## [Editor Report · Acceptance letter]

2 Feb 2023

PONE-D-22-19497R1 

Heat-related mortality in U.S. state and private prisons: a case-crossover analysis 

Dear Dr. Skarha:

I'm pleased to inform you that your manuscript has been deemed suitable for publication in PLOS ONE. Congratulations! Your manuscript is now with our production department. 

Kind regards, 

on behalf of

Dr. Byung Chul Chun 

Academic Editor

PLOS ONE